

# Unsupervised AI reveals insect species-specific genome signatures

Yui Sawada, Ryuhei Minei, Hiromasa Tabata, Toshimichi Ikemura, Kennosuke Wada, Yoshiko Wada, Hiroshi Nagata and Yuki Iwasaki

Department of Bioscience, Nagahama Institute of Bio-Science and Technology, Nagahama-shi, Tamura-cho, Japan

Corresponding authors
Toshimichi Ikemura,
t_ikemura@nagahama-i-bio.ac.jp
Yuki Iwasaki, y_iwasaki@nagahama-i-bio.ac.jp

## ABSTRACT

Insects are a highly diverse phylogeny and possess a wide variety of traits, including the presence or absence of wings and metamorphosis. These diverse traits are of great interest for studying genome evolution, and numerous comparative genomic studies have examined a wide phylogenetic range of insects. Here, we analyzed 22 insects belonging to a wide phylogenetic range (Endopterygota, Paraneoptera, Polyneoptera, Palaeoptera, and other insects) by using a batch-learning self-organizing map (BLSOM) for oligonucleotide compositions in their genomic fragments (100-kb or 1-Mb sequences), which is an unsupervised machine learning algorithm that can extract species-specific characteristics of the oligonucleotide compositions (genome signatures). The genome signature is of particular interest in terms of the mechanisms and biological significance that have caused the species-specific difference, and can be used as a powerful search needle to explore the various roles of genome sequences other than protein coding, and can be used to unveil mysteries hidden in the genome sequence. Since BLSOM is an unsupervised clustering method, the clustering of sequences was performed based on the oligonucleotide composition alone, without providing information about the species from which each fragment sequence was derived. Therefore, not only the interspecies separation, but also the intraspecies separation can be achieved. Here, we have revealed the specific genomic regions with oligonucleotide compositions distinct from the usual sequences of each insect genome, *e.g.*, Mb-level structures found for a grasshopper *Schistocerca americana*. One aim of this study was to compare the genome characteristics of insects with those of vertebrates, especially humans, which are phylogenetically distant from insects. Recently, humans seem to be the "model organism" for which a large amount of information has been accumulated using a variety of cutting-edge and high-throughput technologies. Therefore, it is reasonable to use the abundant information from humans to study insect lineages. The specific regions of Mb length with distinct oligonucleotide compositions have also been previously observed in the human genome. These regions were enriched by transcription factor binding motifs (TFBSs) and hypothesized to be involved in the three-dimensional arrangement of chromosomal DNA in interphase nuclei. The present study characterized the species-specific oligonucleotide compositions (*i.e.*, genome signatures) in insect genomes and identified specific genomic regions with distinct oligonucleotide compositions.

# INTRODUCTION

Insect species have an abundant phylogeny compared to other phyla, with estimates indicating 5.5 million or more species (*Stork, 2018*). Trait diversity in insects is extremely high and numerous variations have been observed, such as the presence or absence of wings and differences in shape, making them highly interesting for comparative genomic studies (*Misof et al., 2014*). Some insects are known to be closely associated with agriculture and human health; therefore, genome sequencing of insects associated with such issues is actively underway (*Li et al., 2019*). Additionally, with the development of long sequencing technologies such as the Pacific Biosciences sequencer, draft genomes of various insects with a high degree of completion are being released (*Li et al., 2019*). However, it should also be noted that due to various problems, such as a high degree of polymorphism in the sequence assembly, it is often difficult to complete their chromosomes; thus, only scaffolds are created (*Richards & Murali, 2015*). Given the large amount of sequence information accumulated on insect genomes, albeit at different levels of completeness, a comparative genomic study on a wide phylogenetic range of insects that is conducted using a method suitable for analyzing large datasets is of particular interest.

A phylogenetic tree analysis based on sequence alignment is undoubtedly a well-established and irreplaceable method for comparative genomic and molecular evolutionary studies (*Kumar et al., 2018*). Notably, sequence alignment-free studies, such as codon usage studies on gene sequences (*Ikemura, 1985*; *Sharp & Li, 1987*; *Duret, 2002*), have also provided important insights into the evolutionary aspect of gene sequences. The present study is a sequence alignment-free study focusing on oligonucleotide usage in genome sequences. Genomic and evolutionary studies have entered the era of big data, and artificial intelligence (AI) technologies suitable for big data analyses may complement conventional phylogenetic methods by providing new perspectives using a large amount of sequence data. The unsupervised machine learning such as SOM (*Kohonen, 1990*) can be used without special models or presumptions and has powerful visualization capabilities, enabling efficient knowledge discovery from a large amount of sequence data (*Kanaya et al., 2001*; *Abe et al., 2003*; *Gatherer, 2007*; *Ikemura et al., 2021*). Importantly, the sequence-alignment free method is less dependent on the completeness of genome sequencing and gene annotation. Other alignment-free methods focusing on the oligonucleotide usage have also been used for genome studies of various species (*Kantorovitz, Robinson & Sinha, 2007*; *Ounit et al., 2015*; *Ahlgren et al., 2017*).

*Karlin, Campbell & Mrázek (1998)* showed that the composition of oligonucleotides varies even among species with the same G + C%, while designating the species-specific composition as the "genome signature". When each genome is fragmented (*e.g.*, 100 kb), the oligonucleotide composition in a majority of fragments is fairly similar, which allows for comparative genomics from a genome signature perspective. Additionally, as oligonucleotide composition can be handled as numerical data, it can be easily inputted into various statistical analyses and AIs. Unsupervised AI can extract unexpected insights from big data without prior knowledge or specific models and is considered highly desirable for current comparative genomics.

We previously developed a batch-learning self-organizing map (BLSOM) for oligonucleotide compositions, which can reveal various novel genome characteristics (*Abe et al., 2003*); the oligonucleotide BLSOM allows the separation (self-organization) of genomic fragments (*e.g.*, 10- and 100-kb) by species and phylogeny without prior information about species during the learning process (*Abe et al., 2005*; *Abe et al., 2006*; *Ikemura et al., 2021*). In the case of higher vertebrates, this unsupervised AI not only classifies (self-organizes) fragment sequences by species, but also reveals the compositional diversity within one genome (*Iwasaki et al., 2013*; *Iwasaki et al., 2022*; *Wada et al., 2015*; *Wada, Wada & Ikemura, 2020*). Importantly, BLSOM is an explainable-type AI that can reveal the diagnostic oligonucleotides that contribute to the classification of genomic sequences.

Human genome research has made great progress with the introduction of various advanced experimental techniques, and humans have become a "model organism" for which a large amount of information has been accumulated. Therefore, it is reasonable to utilize such abundant and diverse knowledge for insect lineage research. Here, the oligonucleotide BLSOM is used for comparative genomic studies of 22 insect species belonging to five phylogenetic groups: Endopterygota, Paraneoptera, Polyneoptera, Palaeoptera, and other insects. The goal of this study is to conduct comparative genomic studies not only among insects from a wide phylogenetic range, but also with a wide range of vertebrates, especially humans, that were previously analyzed using BLSOM: humans (*Iwasaki et al., 2013*; *Wada et al., 2015*; *Wada, Wada & Ikemura, 2020*; *Ikemura et al., 2023*), frogs (*Katsura et al., 2021*), fishes (*Iwasaki et al., 2014*) and bats (*Iwasaki et al., 2022*).

Next, we explain the basic principles and characteristics of this study using unsupervised AI. Although most studies in the biological sciences are based on models and hypotheses that are usually constructed first and then tested, in the present study, we allowed the unsupervised AI to do much of the knowledge discovery without specifying particular models or hypotheses. Then, by focusing on the unexpected and/or characteristic results obtained, we circumstantially investigated the results using more direct methods (*Ikemura et al., 2021*). Although this method is somewhat different from the standard research strategy used in the biological sciences, it can be widely used in fields where large datasets are available: a data-driven science. Overall, the purpose of this study is to introduce this new analysis approach for insect genome studies.

## MATERIALS & METHODS

### Genomic sequences

NCBI has registered the genome sequences of insects classified into a total of 22 orders. The main aim of this study was to compare genomes from a wide range of insect phylogenies, including less-studied phyla. Therefore, we selected one species per order from 22 orders. We choose species with genomic sequences registered by chromosomes or a small number of scaffolds to better understand the overall characteristics of each chromosome and genome. Accordingly, we obtained genome sequences of eight Endopterygota (*Chrysoperla*

*carnea, Mengenilla moldrzyki, Ctenocephalides felis, Drosophila melanogaster, Aricia agestis, Limnephilus lunatus, Aphidius gifuensis*, and *Harmonia axyridis*), four Paraneoptera (*Columbicola columbae, Pediculus humanus corporis, Schizaphis graminum*, and *Thrips palmi*), six Polyneoptera (*Brachyptera putata, Schistocerca americana, Timema cristinae, Anisolabis maritima, Periplaneta americana*, and *Reticulitermes speratus*), two Palaeoptera (*Ischnura elegans* and *Cloeon dipterum*), and two other insects (*Folsomia candida* and *Campodea augens*) from NCBI. Detailed information regarding the accession numbers of each genomic sequence is listed in Table 1 and Table S1. To cover a wide range of phylogenies, genomes with relatively short contig and scaffold sequences were also included in the BLSOM analysis. In such cases, simple analyses only of 1-Mb sequences resulted in the loss of numerous scaffold sequences; thus, after adding N (undetermined nucleotide) to their termini, sequences were concatenated and used for the analysis. Through this N-addition, artificial oligonucleotides that were generated through the concatenation could be excluded from the BLSOM analysis because the oligonucleotides containing N were omitted from the analysis. Although the N addition could not prevent occurrences of artificially concatenated 1-Mb sequences, the 100-kb sequences analyses were also performed later. BLSOM patterns with high similarity were obtained for 1-Mb and 100-kb sequences, supporting the presence of the genome signature in most fragment sequences of a single species. This workflow is illustrated in Fig. S1.

## BLSOM

The Kohonen self-organizing map (SOM), an unsupervised neural network algorithm, is a powerful tool for clustering and visualizing high-dimensional complex data in a two-dimensional map (*Kohonen, 1990*). We modified the conventional SOM for genome informatics based on batch learning such that the learning process and the resulting map were independent of the data input order (*Kanaya et al., 2001*). The initial weight vectors were defined using principal component analysis rather than random values. The weight vectors (wij) were arranged in a two-dimensional lattice denoted by $i$ ($i = 0, 1, \ldots, I - 1$) as well as $j$ ($j = 0, 1, \ldots, J - 1$) and were set and updated as previously described (*Kanaya et al., 2001*; *Abe et al., 2003*). The weights of the first dimension (I) were placed on a lattice corresponding to a width of five times the standard deviation (SD) ($5 \times \sigma 1$) of the first principal component, and the those of lattices in the second dimension (J) were placed on a lattice corresponding to a width of five times the standard deviation (SD) ($5 \times \sigma 2$) of the second principal component. The $\sigma 1$ and $\sigma 2$ were the standard deviations of the first and second principal components, respectively. In the learning step, each genomic fragment was assigned to a lattice point (node) with a weight vector of oligonucleotide composition that was most similar to that of each genomic fragment (the closest Euclidean distance). Then, the value of the weight vector for each node was gradually updated to ensures that it approaches the value of the composition in sequences attributed to the neighborhood of the focused node. This process was repeated several hundred times. The detailed algorithm of BLSOM is described in *Abe et al. (2003)*. The contribution level of each oligonucleotide at each node was visualized using a red/blue heatmap (*Abe et al., 2003*; *Abe et al., 2005*; *Abe et al., 2006*) with dark red, red, white, blue, and dark blue indicating very high, high,

**Table 1** The list of insect species used in this study.

| Order | Organism name | ID | Size (Mb) | G + C% |
|---|---|---|---|---|
| Diplura | *Campodea augens* | GCA_009757345.1 | 1,127.0 | 30.2 |
| Collembola | *Folsomia candida* | GCA_020920555.1 | 219.1 | 37.5 |
| Odonata | *Ischnura elegans* | GCA_921293095.1 | 1,722.7 | 38.5 |
| Ephemeroptera | *Cloeon dipterum* | GCA_902829235.1 | 180.3 | 39.9 |
| Plecoptera | *Brachyptera putata* | GCA_907164805.1 | 436.5 | 36.5 |
| Orthoptera | *Schistocerca americana* | GCA_021461395.2 | 8,990.4 | 42.3 |
| Phasmida | *Timema cristinae* | GCA_002928295.1 | 955.5 | 34.9 |
| Dermaptera | *Anisolabis maritima* | GCA_010014785.1 | 641.7 | 31.1 |
| Blattaria | *Periplaneta americana* | GCA_002939525.1 | 3374.8 | 34.1 |
| Isoptera | *Reticulitermes speratus* | GCA_021186555.1 | 880.6 | 39.9 |
| Mallophaga | *Columbicola columbae* | GCA_016920875.1 | 207.9 | 36.1 |
| Anoplura | *Pediculus humanus corporis* | GCA_000006295.1 | 110.8 | 27.9 |
| Hemiptera | *Schizaphis graminum* | GCA_020882235.1 | 499.2 | 28.0 |
| Thysanoptera | *Thrips palmi* | GCA_012932325.1 | 237.8 | 54.1 |
| Neuroptera | *Chrysoperla carnea* | GCA_905475395.1 | 560.2 | 29.1 |
| Strepsiptera | *Mengenilla moldrzyki* | GCA_000281935.1 | 155.7 | 28.7 |
| Siphonaptera | *Ctenocephalides felis* | GCA_003426905.1 | 775.5 | 29.8 |
| Diptera | *Drosophila melanogaster* | GCA_002310775.1 | 120.4 | 42.4 |
| Lepidoptera | *Aricia agestis* | GCA_905147365.1 | 435.3 | 36.9 |
| Trichoptera | *Limnephilus lunatus* | GCA_917563855.2 | 1,269.7 | 35.0 |
| Hymenoptera | *Aphidius gifuensis* | GCA_014905175.1 | 157.0 | 26.5 |
| Coleoptera | *Harmonia axyridis* | GCA_914767665.1 | 425.5 | 35.1 |

moderate, low, and very low contributions, respectively. The current study focused on vectorial data consisting of the oligonucleotide frequency at each node; we divided the range between the maximum and minimum frequencies of each oligonucleotide into 21 and then colored each node by the division to which the frequency of the node belonged. Accordingly, the dark red (blue) area of the red/blue heatmap indicates sequences in which the frequency of the focused oligonucleotide was distinctly higher (lower) than that of other sequences. Orange/blue heatmaps were also presented for those with non-normal color vision; see figure legends. The BLSOM and oligonucleotide-count program can be obtained from a GitHub repository (https://doi.org/10.6084/m9.figshare.25036358.v1).

## RESULTS

### BLSOM for 1-Mb sequences of 22 insect genomes

To elucidate and compare genome characteristics of insects belonging to a wide phylogenic range, we conducted BLSOM analyses of oligonucleotide compositions firstly of 1-Mb sequences, derived from 22 insect genomes (Table 1). Genomic sequences registered in public DNA databases (*e.g.*, International Nucleotide Sequence Database Collaboration; https://www.insdc.org/about) are one of two complementary strands, and in the registration of genomic sequences, especially in the case of scaffolded sequences, one

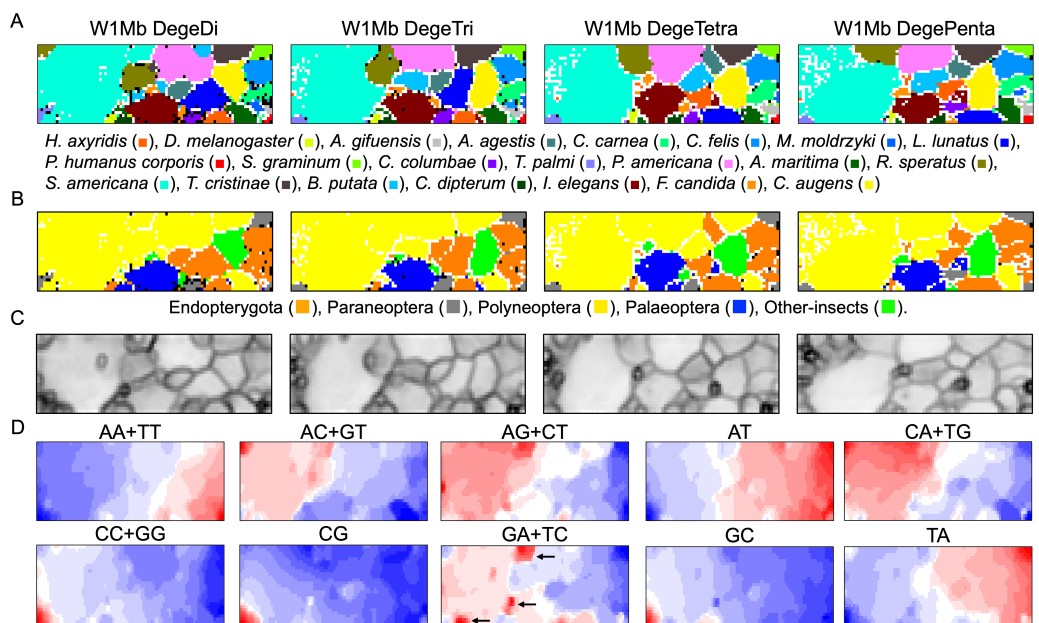

**Figure 1 Oligonucleotide BLSOM for 1-Mb sequences.** (A) DegeDi, DegeTri, DegeTetra, and DegePenta BLSOM for 22 insect genomes. Nodes containing sequences from multiple species are indicated in black and those containing sequences from a single species are colored as shown below the figure. Nodes that include no sequences were left blank (vacant). Territories of each insect on W1Mb BLSOMs are presented in Figs. S2–S5. (B) Nodes containing sequences from multiple phylogenetic lineages are indicated in black and those containing sequences from a single lineage are colored as shown below the figure. (C) U-matrix of each BLSOM. (D) Heatmap of each BLSOM. Contribution of each oligonucleotide at each node is visualized by color: dark red (very high), red (high), white (moderate), blue (low), and dark blue (very low). Other examples of heatmaps of W1Mb BLSOMs are presented in Fig. S6. Orange/blue heatmap patterns were also presented, for the easy accessibility to those with non-normal color vision as Fig. S14.

of the complementary sequences is selected quite arbitrarily; *i.e.*, the distinction between complementary sequences is usually not important for understanding global properties of genomic sequences such as genomic signatures. In this study, two complementary oligonucleotides (*e.g.*, AA and TT) were added together, and the oligonucleotide sets were referred as degenerate di-, tri-, tetra-, and penta-nucleotides (DegeDi, DegeTri, DegeTetra, and DegePenta) (*Abe et al., 2005*; *Abe et al., 2006*). Figure 1A illustrates the BLSOM for the compositions (%) of these oligonucleotides in all 1-Mb fragments derived from 22 insect genomes (W1Mb BLSOM). The number of grid points (nodes) was set as such that an average of 10 sequences was attributed to each node.

As BLSOM is an unsupervised machine learning algorithm, only data of oligonucleotide compositions in fragment sequences were given during the learning step, and the sequences with close Euclidean distances were clustered (self-organized) on the map. In the figure, nodes containing sequences of a single species are shown in species-specific colors, and those containing sequences of more than one species are displayed in black. Even with no information concerning species during the learning process, species-dependent clustering (self-organization) of fragment sequences is clear, *i.e.*, sequences from individual species form their own territories. Although the number of nodes is set so that each node contains

an average of 10 sequences, the number of black nodes is very small in all four maps. This shows that the clustering power of the oligonucleotide BLSOM is very high and the unique genome signature extends over almost the entire region of each insect genome. More specifically, the number of black nodes on DegeTetra and DegePenta was slightly smaller than those on DegeDi and DegeTri, indicating the slightly increased clustering power of longer oligonucleotides compared to those of shorter oligonucleotides (Fig. 1A).

Twenty-two species were classified into the following five phylogenetic groups and then colored accordingly: Endopterygota, Paraneoptera, Polyneoptera, Palaeoptera, and Other-insects (Fig. 1B). On each BLSOM in Fig. 1B, most sequences of species belonging to Polyneoptera (yellow) formed a continuous large territory on the left. In addition, sequences of the species belonging to Endopterygota (Indian yellow) tended to form mutually adjacent regions in the lower side, and a similar tendency was observed for Palaeoptera (blue). Alternatively, for Paraneoptera (ash gray) and Other-insects (lime) sequences of individual species tended to form isolated territories. This suggests that species phylogenetically close to each other tend to come into close proximity on the BLSOM. Based on the black node in this phylogenetic grouping (Fig. 1B), DegePenta appears to have the best resolution.

BLSOM is equipped with a tool known as the U-matrix (*Ultsch, 1993*), which displays the Euclidean distance between representative vectors of neighboring nodes as the degree of blackness, with larger distances being reflected by a higher degree of blackness (Fig. 1C). When comparing the territory of each species represented in Fig. 1A, boundaries of species territories mostly correspond to the clear black lines of the U-matrix (Fig. 1C); thus, showing that the oligonucleotide composition clearly differs depending on the species. Additionally, for the largest *Schistocerca americana* territory (colored in cobalt green in Fig. 1A), there are clear black lines of the U-matrix even within this species territory, showing that the Mb-level sequences with distinct oligonucleotide compositions seem to coexist in this grasshopper genome. In the left and mainly upper part in the *S. americana* territory, especially on the DegeTetra and DegePenta BLSOM, there are vacant (colorless) nodes that do not contain any sequences (Fig. 1A). For the node to which sequences with distinct oligonucleotide composition from others are attributed, no sequence tends to be attributed to the nodes adjacent to the focused node after the learning, resulting in vacant nodes (*Iwasaki et al., 2013*; *Wada et al., 2015*). This shows that sequences located in the left and mainly upper part in the *S. americana* territory should differ distinctly in their oligonucleotide composition, not only from a majority of other *S. americana* sequences, but also from each other: the subterritories of this grasshopper.

## Dinucleotides contributing species-dependent clustering

BLSOM is an explainable-type machine learning that provides the reason for sequence clustering (self-organization) using a red/blue heatmap (Fig. 1D) (*Abe et al., 2003*; *Abe et al., 2005*; *Abe et al., 2006*). The contribution of each oligonucleotide at each node can be visualized based on color: dark red (very high), red (high), white (moderate), blue (low), and dark blue (very low). Figure 1D shows the heatmaps for DegeDi BLSOM (Fig. 1A). Sequences with a high content of dinucleotides consisting of only A or T occupy a large

area, and those with a high content of dinucleotides consisting of only C or G occupy only a small territory in the lower left. This indicates that the insect genomes are mainly (G + C)-poor, while some are (G + C)-rich. The small (G + C)-rich territory contained a majority of *T. palmi* sequences and a small portion of *S. americana* sequences, supporting the findings of a previous study (*Rotenberg et al., 2020*) in which the genome of *T. palmi* was unusually (G + C)-rich (50.9 G + C%) compared to other insect genomes.

To further clarify the results obtained using DegeDi BLSOM, we performed a box plot analysis of the dinucleotide percentage along with the G + C% in each 1-Mb fragment (Fig. 2); see its legend for details. First, the results of G + C% (Fig. 2A) will be explained. The entire box of G + C% for *T. palmi* was located above 50%, *i.e.*, G + C% of most 1-Mb sequences is more than 50%, which supports the result reported by *Rotenberg et al. (2020)*. For *S. americana*, while the entire box for the G + C% and its median were less than 50%, numerous outliers were located above 50%. For *A. agestis*, most outliers were on the (G + C)-rich side, while for *I. elegans* they were primarily on the (A + T)-rich side. When focusing on *A. gifuensis, C. carnea,* and *S. graminum*, we observed low genome G + C%. Only six chromosome sequences of these three species were registered in the NCBI; the smallest number of chromosomes among the 12 species whose genomes have been determined at the chromosome level. To further examine the relationship between the number of chromosomes and the genomic and chromosomal G + C%, we calculated the G + C% for each chromosome of the 12 species (Fig. 2B). The G + C% for each chromosome of the above three species is lower than 35% and lower than those of most other insects with seven or more chromosomes.

Figures 2C and 2D show the results of box plot analysis of CG% and GC%. Naturally, species with high genome G + C% also have high values of these contents, while species with low genomic G + C% have low contents. Comparison between the two dinucleotides shows a high degree of similarity, but as indicated on the vertical axis, the CG% tends to be lower than the GC% for many species. When analyzing the CG/GC ratio in detail, ratios below 1, and thus CG-deficiencies, were found in most insects, with clear exceptions of species belonging to Paraneoptera (Fig. S8B). The marked CG deficiency in mammalian genomes is known to be related to the C-methylation of CG (*Bogdanovic & Veenstra, 2009*). However, in insects the CG deficiency is known to be clearly less than that in mammals (*Bewick et al., 2017*). The CG deficiency is commonly expressed as an odds ratio of CG frequency (Observed /Expected), and therefore this value is shown in Fig. 3 for 22 insects. Here, insects are grouped by phylum, and species belonging to Polyneoptera tend to have the most pronounced CG deficiency, followed by species belonging to Endopterygota. As mentioned in the CG/GC ratio, Paraneoptera species tend to have an excess of CG, which is different from most of other insects. This general trend among insect lineages should provide insight into the evolutionary processes of methylation and epigenetic mechanisms in the insect lineage (*Bewick et al., 2017*).

Regarding other dinucleotides, AC + GT, AG + CT, CA + TG, and GA + TC were enriched (red in Fig. 1D) mainly in the large *S. americana* territory. Especially about GA + TC, three dark red (evidently enriched) zones were observed and arrowed, which were separated from the major *S. americana* territory by dark black lines in the U-matrix; this

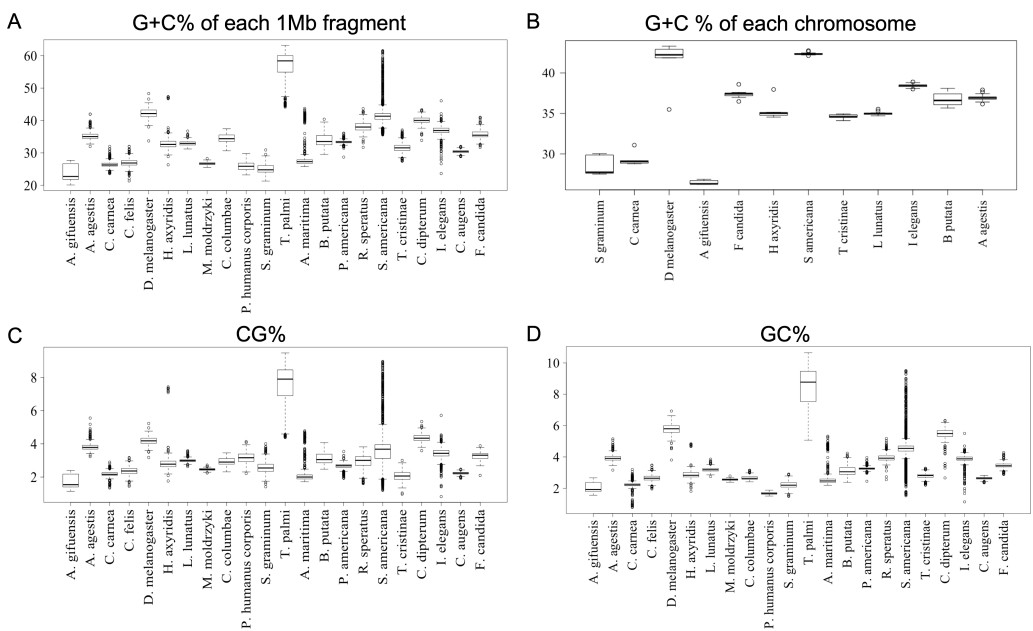

**Figure 2  Boxplot of G + C% and dinucleotide %.** Boxplots showing occurrence frequencies (%) of G + C or dinucleotides. The boxplot was generated using the boxplot function in R. The maximum length of the whiskers in the boxplot is set to a default value of 1.5, and any data points that exceed this limit are considered outliers. G + C% in each 1 Mb fragment (A) or each chromosome (B) of each insect. CG% (C) and GC% (D) in 1 Mb fragments. Boxplots of other dinucleotides are presented in Fig. S8.

indicates that Mb-level structures are distinctly enriched with GA + TC in its genome. In the GA + TC arrowed areas, some increase in red color was also observed for AG + CT, but to a lesser extent. We set one target of this study to understand the biological significance of the characteristics of GA + TC for the following reasons. In our previous dinucleotide BLSOM analysis of the human genome, GA + TC gave also the most distinctive heatmap pattern among the dinucleotides, and detailed analysis has revealed that the distinctive increase in red color is because the GA + TC frequency is significantly enriched in the centromeric and pericentromeric heterochromatin region (*Iwasaki et al., 2022*).

## Chromosomal distribution of occurrence frequencies of dinucleotides

The genome signature is of particular interest in terms of the mechanisms and biological significance that have caused the species-specific difference, and is one of the powerful search needles to explore the various roles of genome sequences other than protein coding, and can be used as targets to solve the many mysteries hidden in the genome sequence. In this study, the initial data mining was performed using BLSOM, and when this unsupervised AI yielded unexpected and/or characteristic results, the results were circumstantially investigated using more direct methods. To learn more about the findings illustrated in Fig. 1, we focused first on *S. americana* as a representative example and analyzed the distribution of dinucleotide occurrence frequencies on each chromosome of this grasshopper species. In Fig. 4, we plotted the distribution patterns of occurrence frequencies (%) in 1-Mb sequences for four dinucleotides, including GA + TC, on

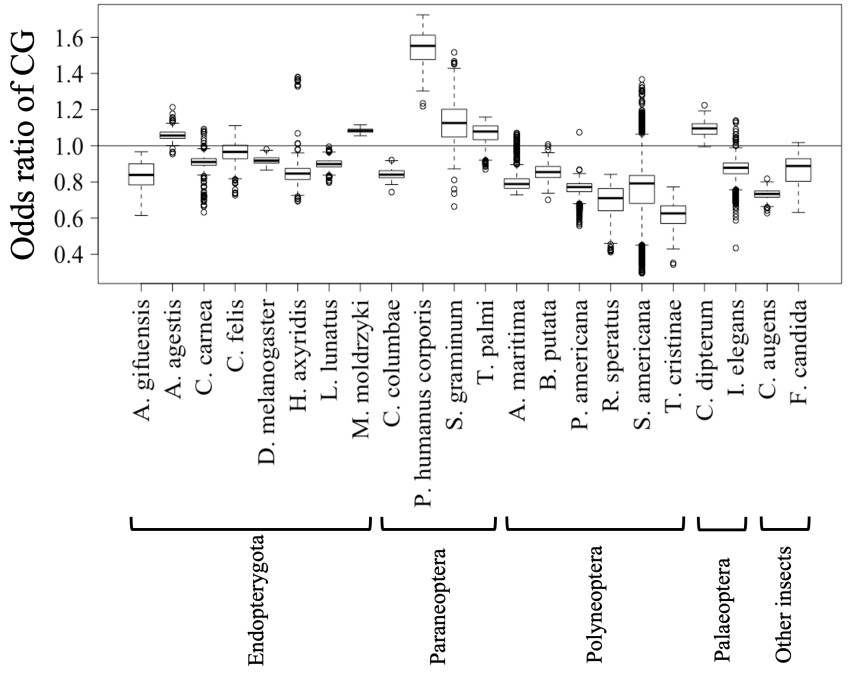

**Figure 3** **Boxplot of odds ratio of dinucleotides.** Boxplots showing the odds ratio (observed/expected) of CG in each 1 Mb fragment. The expected frequency of each dinucleotide was calculated from the mononucleotide composition at each fragment. The boxplot was generated using the same method as described in Fig. 2. Insects are grouped into five phylotypes as shown in Fig. 1B.

chromosomes 1–4, while examples for other chromosomes are shown in Fig. S9. For the upper two dinucleotides, numerous data points have higher values than those of the basal level, therefore forming distinct peaks in a region spanning several tens of Mb at and near the chromosome end (Fig. 4). Importantly, for these dinucleotides, similar distinctive peaks were observed at the end of all other *S. americana* chromosomes and the GA + TC peak was more pronounced than the AG + CT peak for all chromosomes, which suggests that the former, prominent peak is not a mere reflection of the increase in the composition of mononucleotides making up these dinucleotides. Features identified in almost all chromosomes of a given species are considered species-specific and provide incentives to study their biological significance. Karyotype analyses of *Schistocerca* species have shown that their chromosomes are composed exclusively of acrocentric chromosomes and that their centromeric and pericentromeric regions are located in the terminal region of their chromosomes (*Souza & Melo, 2007*). This indicates that most *S. americana* chromosomes are also acrocentric and that the GA + TC peaks in their terminal regions (Fig. 4) are related to the specific oligonucleotide compositions in their centromeric and pericentromeric regions. It should again be noted that the evident GA + TC peaks were found in the centromeric and pericentromeric regions of human chromosomes (*Wada et al., 2015*; *Wada, Wada & Ikemura, 2020*). This similarity between distantly phylogenetically related species is unexpected and intriguing. As a reference, the lower side of Fig. 4 shows results of CG and GC, with patterns that are significantly different from those of GA + TC.

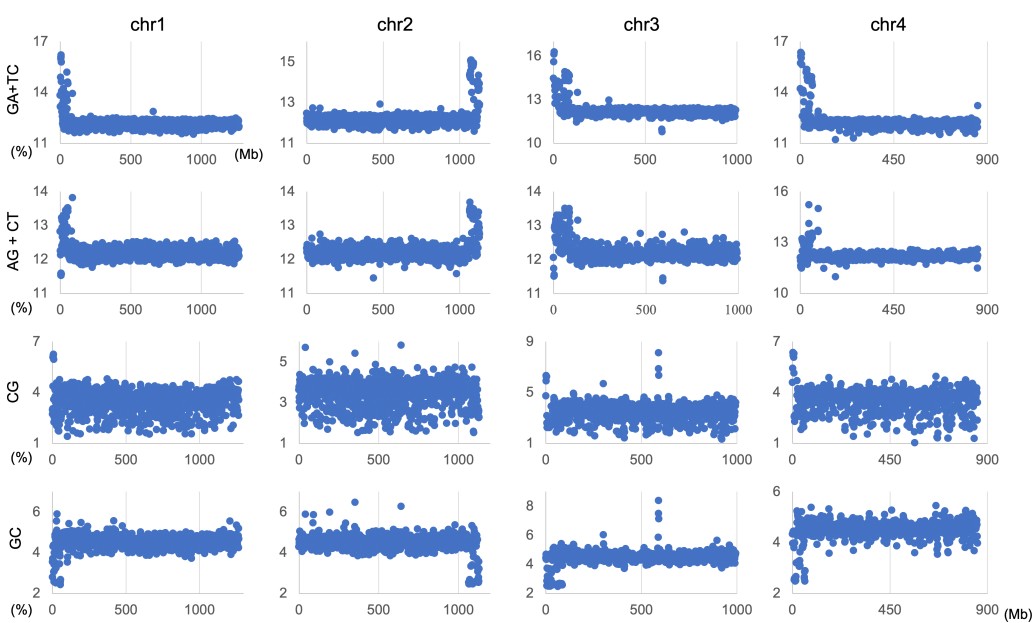

**Figure 4** **Chromosomal distribution of dinucleotides.** Distribution charts showing chromosomal distribution of GA + TC, AG + CT, CG, or GC on chromosomes 1, 2, 3, and 4 in 1-Mb sequences of *S. americana* are presented. The vertical axis represents the occurrence frequency (%) of each dinucleotide. Distribution charts on other chromosomes are presented in Fig. S9.

Although clear peaks (as in the above two dinucleotide pairs) were not seen, small peaks were observable for CG at one end, except in chromosome 2. Although the peak seen at the end is not seen in GC, a downward peak is often observed at the end (Fig. S9), indicating that the CG peaks seen at the chromosome ends are not due to a simple reflection of the terminal increase in G + C%.

## Mb-level structures with distinct trinucleotide compositions

BLSOM and box plots of dinucleotide compositions and their distribution on *S. americana* chromosomes revealed Mb-level structures with distinct dinucleotide compositions, and the clearest example was the evident enrichment in GA + TC in terminal regions of all *S. americana* chromosomes. By focusing on DegeTri BLSOM (Fig. 5A), we further characterized the Mb-level structures with distinct trinucleotide compositions, first in the *S. americana* genome and then in other species genomes. Eight heatmap patterns for trinucleotide pairs containing GA and TC are shown in Fig. 5B. Patterns of all trinucleotide pairs are shown in Fig. S10. The dark red in the heatmap is mainly seen in areas corresponding to *S. americana* sub-regions at the left end, which are surrounded by black lines in the U-matrix (the lower panel in Fig. 5A). Notably, the location of the dark red zones depends on the type of mononucleotide added to GA + TC (Fig. 5B); *i.e.,* depending on the nucleotide added to GA and TC, different trinucleotides were differentially enriched in the focused sub-territories. Even outside the *S. americana* territory, spot-type dark red areas were observed for several trinucleotides (*e.g.,* AGA + TCT, ATC + GAT, CTC +

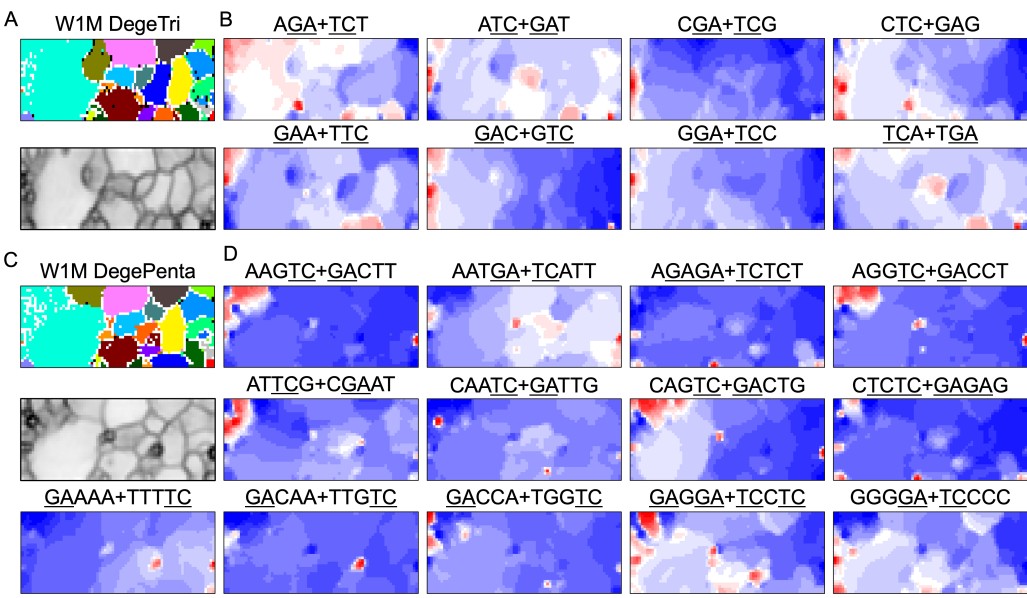

**Figure 5** **BLSOMs for DegeTri and DegePenta compositions.** (A and B) DegeTri and (C and D) Dege-Penta BLSOM. Heatmaps showing the contributions of DegeTri- and DegePenta containing GA + TC are presented (B and D, respectively); GA and TC sequences are underlined. Heatmaps for other trinucleotides are presented in Fig. S10. Orange/blue heatmap patterns were also presented, for the easy accessibility to those with non-normal color vision as Fig. S15.

GAG, GAC + GTC, and TCA + TGA), indicating that characteristic sequences with these specific trinucleotide compositions also exist in other insects.

### Mb-level structures with distinct pentanucleotide compositions

When considering tetranucleotides and longer oligonucleotides, cores of DNA-binding protein motif sequences (*e.g.*, transcription factor binding sites; TFBSs) are included. This facilitates the consideration of oligonucleotide composition from a biological perspective. All 512 heatmaps of pentanucleotides are shown in Data S1, while Fig. 5D illustrates cases including GA and TC, which showed characteristic dark red spots in the heatmap of di- and trinucleotide BLSOM (Figs. 1D and 5B). Dark red zones were observed in sub-territories of *S. americana*, while small, dark red spots were observed in various non-*S. americana* territories, which primarily correspond to the small areas surrounded by dark black lines in the U-matrix. This shows that there are sequences that are apparently enriched in pentanucleotides containing GA and TC also in non-*S. americana* genomes, albeit on a shorter genomic scale. Since the BLSOMs in Fig. 5C are based on pentanucleotide compositions, the special zones of non-*S. americana* species, which are located apart from *S. americana* sub-territories on the map, should enrich different pentanucleotide sets compared to those from the *S. americana* genome.

### Dinucleotide BLSOM and distribution analyses for 100-kb sequences

By analyzing all 1-Mb sequences derived from each genome, we identified species-specific oligonucleotide compositions (genome signatures), which were observed in almost the

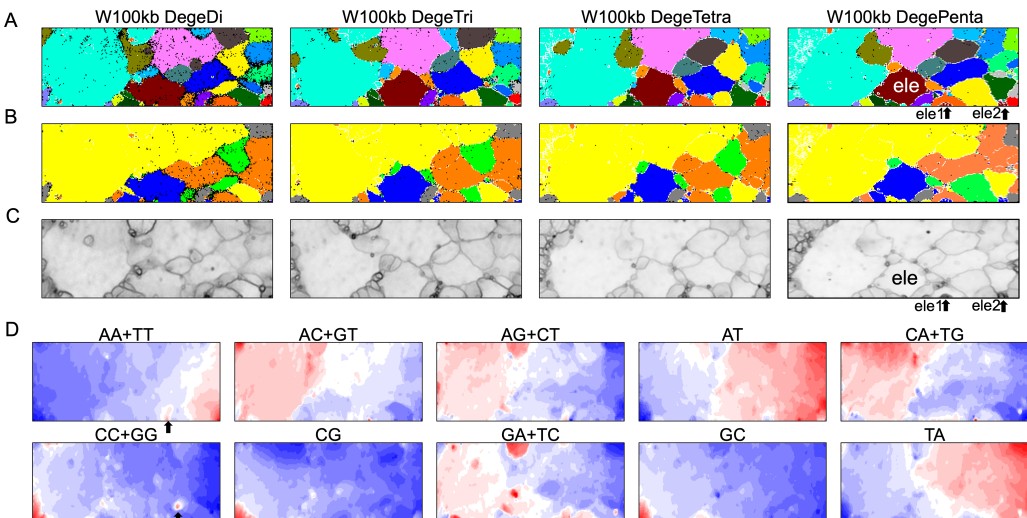

**Figure 6** **BLSOMs for 100-kb sequences.** (A) Nodes containing sequences from multiple species are indicated in black and those containing sequences from a single species are colored as described in Fig. 1A. The territories of each insect are presented in Fig. S7. (B) Nodes containing sequences from multiple lineages are indicated in black and those containing sequences from a single lineage are colored as described in Fig. 1B. (C) U-matrix of each BLSOM. (D) Heatmaps of all dinucleotides for W100kb DegeDi BLSOM. In AA + TT and CC + GG, localized red areas are indicated by arrows. Orange/blue heatmap patterns were also presented, for the easy accessibility to those with non-normal color vision as Fig. S16.

entire region of each genome, as well as the existence of local Mb-level regions with distinct oligonucleotide compositions within a single genome, particularly in the *S. americana* genome. To further explore the special local structures of a species with a much smaller genome than *S. americana*, we performed a BLSOM analysis of 100-kb fragments. The results are illustrated in Fig. 6, indicating again a good separation of species. Figure 6B shows the heatmaps of dinucleotides for the 100-kb DegeDi BLSOM. Their patterns are similar to those observed in Fig. 1D, whereas the dark red areas of GA + TC became more prominent. In addition, for AA + TT and CC + GG, localized red areas (arrowed) were observed within non-*S. americana* territories. To clarify the characteristics of these special sequences, we investigated the distribution of each dinucleotide pair in 100-kb sequences on each chromosome in the species for which relatively long sequences had been registered for several chromosomes. Figure 7 shows the results for chromosomes 1–4 of four species. The first row shows GA + TC results for *S. americana*, the second row shows AA + TT of *I. elegans*, the third row shows the CC + GG results for *F. candida*, and the fourth row shows CG of *B. putata*. Although different patterns were observed for different species, when focusing on a single species, a particular dinucleotide pair showed a similar distribution pattern across different chromosomes; thus, suggesting its biological significance.

## Special zones (SZs) in pentanucleotide BLSOM for 100-kb sequences
In the U-matrix of the W100kb DegePenta BLSOM (Fig. 6C), numerous small zones surrounded by clearer black lines than those in the W1Mb DegePenta BLSOM were observed (Fig. 1C), including those in non-*S. americana* territories. These often form

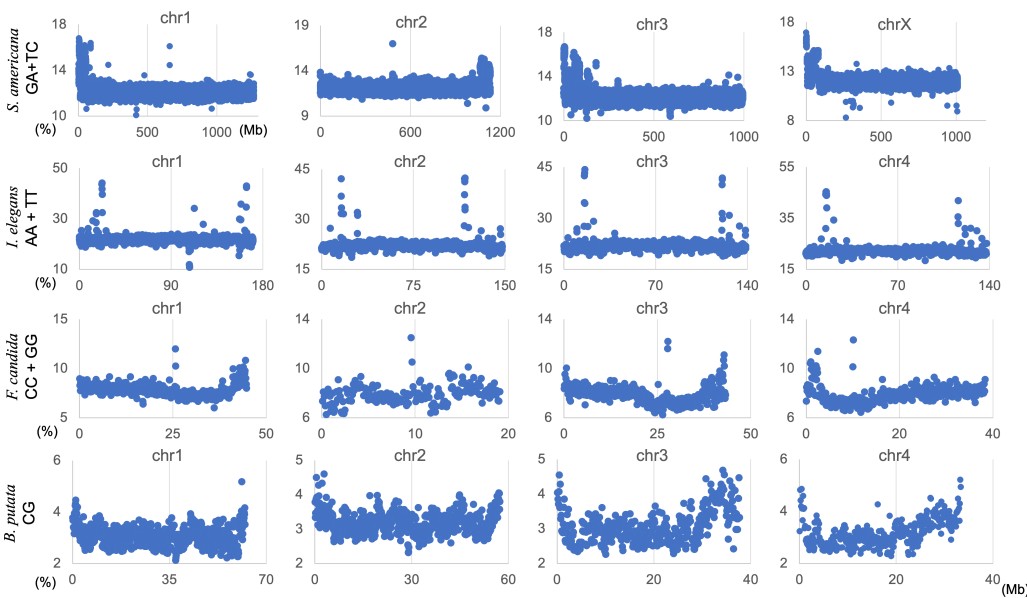

**Figure 7  Chromosomal distribution of DegeDi in 100-kb sequences of four insects.** Distribution charts showing chromosomal distribution of GA + TC for *S. amerinaca*, AA + TT for *I. elegans*, CC + GG for *F. candida*, and CG for *B. putata* on four chromosomes are presented. The vertical axis represents the occurrence frequency (%) of each DegeDi.

satellite territories, which are located away from the major territory of each species and are therefore thought to have oligonucleotide compositions distinct from those of the main territory. For instance, satellite territories of *I. elegans* (ele1 and 2) are shown along with its major territory (ele) in Figs. 6A and 6C. As we will explain in detail later, we have previously identified similar satellite-like zones, which are surrounded by clear black lines of U-matrix in the oligonucleotide BLSOM analysis of humans (*Iwasaki et al., 2013*; *Wada et al., 2015*; *Wada, Wada & Ikemura, 2020*), seven frogs (*Katsura et al., 2021*), and six bats (*Iwasaki et al., 2022*). We named them "special zones" (SZs), and in this study, we named similar zones of insects as insect SZs. The sequences attributed to the SZs are named "SZ sequences". Of note, in non-*S. americana* territories, small SZs were often hard to find in the U-matrix of 1-Mb BLSOM (Fig. 1C), indicating that SZ sequences observed in 100-kb BLSOM are primarily shorter than 1 Mb.

Distribution analyses of pentanucleotides from 100-kb sequences can reveal detailed features of SZs, such as their locations on chromosomes (Fig. 7). In fact, we previously conducted a distribution analysis of penta- and hexanucleotide content on 100-kb or 1-Mb sequences for each human and bat chromosome and identified peaks for a wide variety of oligonucleotides with a markedly elevated frequency (*Wada, Wada & Ikemura, 2020*; *Iwasaki et al., 2022*). Notably, even for pentanucleotides, there are 512 types, and since all chromosomes were treated individually, this standard distribution analysis was inevitably large. For the insect genomes, a similar distribution analysis of pentanucleotides was performed in an exploratory manner, which was made more complicated by the inclusion of multiple species. The results are partially shown in Fig. S11.

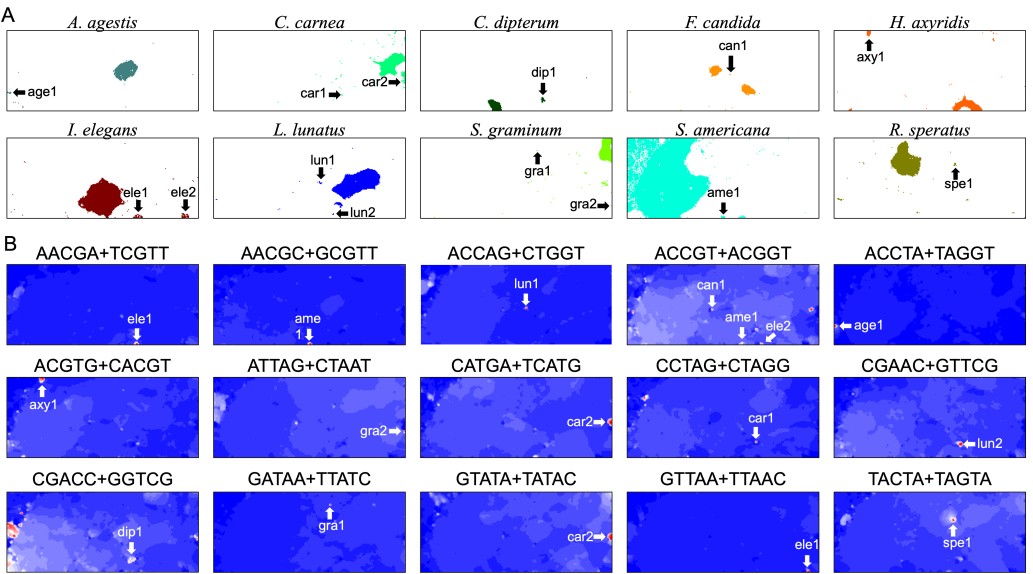

**Figure 8   Insect special zones (SZs).** (A) Insect territories are indicated by color, while SZs are marked with arrows and named; the coloring follows Fig. 1. (B) Heatmap of the DegePenta, which is particularly enriched in SZs. Orange/blue heatmap patterns were also presented, for the easy accessibility to those with non-normal color vision as Fig. S17.

In contrast, for BLSOM, all pentanucleotides of all chromosomes of all target species can be analyzed at once, showing the usefulness of AI technologies. This allows us to study the overall features of genomic sequences with specific oligonucleotide compositions, such as SZ sequences, even for multiple species. In Fig. 8A, territories of each of ten insects are indicated by colors, and SZs are marked with arrows and named. In Fig. 8B, fifteen examples of pentanucleotides that were particularly enriched in these SZs, are shown. Only fifteen examples are shown here, but there are multiple, highly similar heatmap patterns (Data S2), indicating that multiple pentanucleotides are enriched in each SZ. Collectively, a group of oligonucleotides occurs frequently in the 100 kb (or Mb) level SZ sequences, and therefore these sequences are considered to be 100 kb (or Mb) level sequences, possibly with a high concentration of specific repetitive sequences.

## BLSOM for a single species

By analyzing 22 insects at once, we discovered that *S. americana* has numerous SZs while others have a much smaller number of SZs. To further investigate the characteristics and biological significance of each species' SZ, BLSOM for a single species is considered useful. We therefore constructed 100-kb DegePenta BLSOMs for *S. americana*, *I. elegans*, *A. agestis*, and *L. lunatus* separately (Figs. 9A–9D) and compared the obtained patterns with that of the human genome (Fig. 9E). Nodes with sequences from multiple chromosomes are displayed in black, while nodes containing sequences from a single chromosome are indicated with chromosome-specific colors. In the case of *S. americana* (Fig. 9A), many distinct SZs bordered by clear black lines in the U-matrix are clearly visible, and the SZs are often composed of monochromatic nodes, *i.e.,* chromosome-specific SZs. This feature

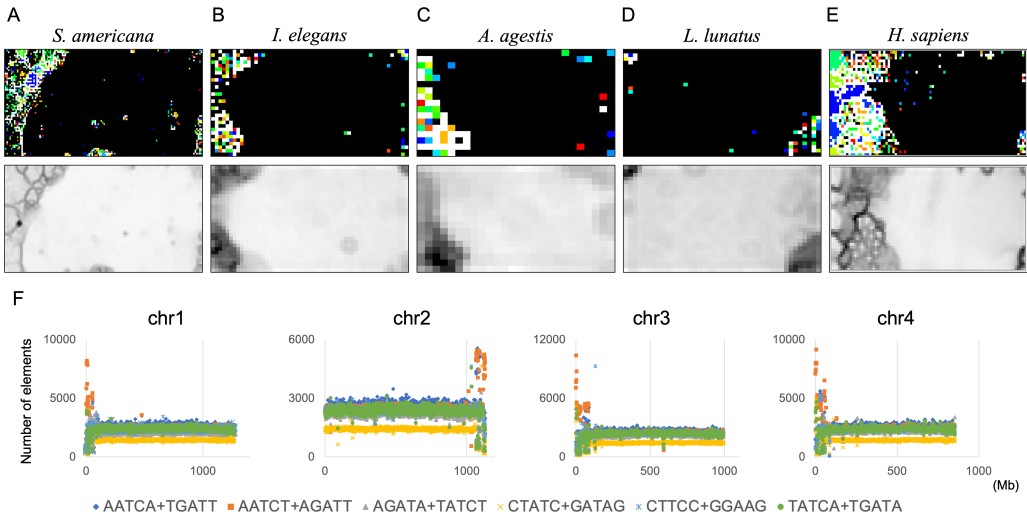

**Figure 9  DegePenta BLSOM for 100-kb sequences of a single species.** (A) *S. americana*. (B) *I. elegans*. (C) *A. agestis*. (D) *L. lunatus*. (E) *H. sapiens* (humans). Nodes containing sequences from multiple chromosomes are indicated in black and those containing sequences from a single chromosome are colored. The color of each chromosome on the BLSOM is shown in Table S4. The U-matrix is displayed under each BLSOM. (F) Distribution charts showing chromosomal distribution of six TFBS consensus core elements for *S. americana* are presented. The vertical axis represents the number of each TFBS consensus core element.

is more similar to that of the human genome (Fig. 9D) than to that of other insects, where the separation by the U-matrix line within the SZ is less clear. The internal separation in SZ is not clear in these insects, but the SZ is clearly separated from their main territory by U-matrix lines, showing the presence of sequences with different oligonucleotide compositions. However, the nodes tend to be scattered in various colors and it is rather difficult to find clear sub-territories composed only of specific colors, which contrasts with the patterns in humans and *S. americana*.

## Comparison with vertebrate genomes

As explained in the Introduction section, the aim of this study is not only the comparison among insects, but also the comparison with other phylogenetic groups that have been well characterized. As noted above, we have characterized the genomes of humans, frogs, and bats, using BLSOM of oligonucleotide compositions and their distribution on each chromosome. Extensive phylogenomic analyses of insects reported by *Misof et al. (2014)* date the origin of insects to the Early Ordovician (∼479 million years ago (Ma)) era, the insect flight to the Early Devonian (∼406 Ma) era, the major extant lineages to the Mississippian (∼345 Ma) era, and the major diversification of holometabolous insects to the Early Cretaceous era. During this long evolutionary process, the diversification of various genome characteristics should have progressed along with the remarkable diversification of phenotypes. Comparison with vertebrates, which are phylogenetically distant from insects, may elucidate various genome characteristics, such as those possibly acquired through convergent evolution.

As noted above, the DegeDi BLSOM heat map of the human genome showed a dark red zone composed of Mb-level sequences with significantly increased GA + TC content (*Iwasaki et al., 2022*), which is similar to the dark red zone observed in the S. americana territory and arrowed in the GA + TC heat map in Fig. 1D. The distribution analysis of GA + TC content in human chromosomes showed that the focused sequences were derived from centromeric and pericentromeric heterochromatin regions (*Iwasaki et al., 2022*), where an obvious enrichment with diverse TFBSs was found, and the genomic regions enriched with diverse TFBSs were called "Mb-level TFBS islands" (*Iwasaki et al., 2013*; *Iwasaki et al., 2022*; *Wada, Wada & Ikemura, 2020*). Notably, a wide range of TFBSs enriched in the human Mb-level TFBS islands were found to frequently contain GAs and TCs in their motif sequences, which was thought to be the cause of the distinct enrichment with the GA + TC dinucleotide in the centromeric and pericentromeric regions (*Wada, Wada & Ikemura, 2020*; *Iwasaki et al., 2022*). TFBS sequences are known to be well conserved during evolution, and using the TFBS sequences identified in humans, we analyzed the bat genomes and discovered regions where diverse TFBSs were enriched in each chromosome, but on a much smaller scale than that observed in the human genome (*Iwasaki et al., 2022*). Combining the results obtained for these vertebrates and the present insects, it seems interesting to consider the relationship between insect SZ sequences and TFBSs.

As the first analysis, we attempted to compare the results of the DegePenta composition of *S. americana* with those obtained for humans. In an aforementioned study (*Iwasaki et al., 2013*), we identified ten DegePenta sequences as consensus core elements of diverse TFBS motifs, referring to JASPAR Core Vertebrata (https://jaspar.elixir.no/) (*Wasserman & Sandelin, 2004*) and TRANSFAC Public (http://www.gene-regulation.com/pub/databases.html) (*Matys et al., 2006*) and showed that the ten sequences are significantly concentrated in human centromeric and pericentromeric heterochromatin regions. To explore the possibility that these TFBS motifs are also concentrated in the corresponding region of *S. americana*, we examined the distribution on each chromosome of the ten sequences used in the human analysis. Interestingly, for the frequencies of the six TFBS consensus motifs, containing GA and TC dinucleotides, prominent peaks were observed at and near the end of all acrocentric chromosomes of this species (Fig. 9B and Fig. S12). This chromosomal location of the peaks coincided with that of the GA + TC peak in Fig. 4. The formation of highly similar megastructures in phylogenetically distant species, which appear to have been acquired in a convergent evolutionary manner, may have biological significance. For the human genome, we have also analyzed a large number of Dege-Hexa and Hepta TFBSs that were obtained from SwissRegulon Portal (http://swissregulon.unibas.ch/sr/downloads) (*Pachkov et al., 2013*), and we plan to do the same for *S. americana*. We are also attempting to analyze small scale SZ sequences of other insects, which appear to be clearly different from those of humans, by using HOMER (*Heinz et al., 2010*), in which a weight matrix of transcription factor binding motifs from various species is obtained. When focusing on the SZ sequences that were identified using U-matrix and visualized in Fig. 8A, various TFBSs were discovered to be enriched in SZ sequences. However, the analysis for these species is

clearly in its early stages compared to that for *S. americana* and is therefore presented as Supplemental Data (Table S3).

## DISCUSSION

### Biological significance of large-scale structures with distinct oligonucleotide compositions

The biological significance of large structures with distinct oligonucleotide compositions, *i.e.,* 100 kb- or Mb-level sequences with a high concentration of specific sequences with distinct oligonucleotide compositions, is first discussed. It should be noted that the non-coding sequences such as repetitive sequences were once thought to be a typical example of junk DNA but are now thought to be associated with diverse functions (*Sasaki et al., 2008*; *Hirakawa et al., 2009*). As mentioned above, BLSOM and distribution analyses of penta- to hexanucleotide compositions from the 1-Mb human sequences revealed an enrichment with diverse TFBSs (Mb-level TFBS islands) in the centromeric and pericentromeric heterochromatin region, and interestingly, the types and combinations of TFBSs enriched in the human Mb-level TFBS islands differed by chromosome (*Wada et al., 2015*; *Wada, Wada & Ikemura, 2020*). The centromere region has been shown to be primarily composed of alpha-satellite sequences, and alpha-satellite monomer sequences differ among chromosomes (*Hayden et al., 2013*; *Aldrup-MacDonald et al., 2016*; *Sullivan, Chew & Sullivan, 2017*). This chromosome-dependent difference in alpha-satellite monomer sequences is thought to explain the existence of chromosome-dependent SZs (*Wada, Wada & Ikemura, 2020*; *Iwasaki et al., 2022*).

A common function of centromeric and pericentromeric regions is the formation of condensed heterochromatin in chromocenters, which supports the association of centromeric and pericentromeric DNAs of homologous, as well as nonhomologous, chromosomes in interphase nuclei (*Maison et al., 2002*; *Maison & Almouzni, 2004*; *Probst, Dunleavy & Almouzni, 2009*; *Probst & Almouzni, 2011*). Recently, functions of transcription factors (TFs) other than their known gene expression-regulatory functions have received much attention (*MacQuarrie et al., 2011*; *Sanyal et al., 2012*). Of these functions, TF-mediated looping interactions between two different genomic regions has become a popular topic (*Dixon et al., 2012*; *Dixon, Gorkin & Ren, 2016*; *Maison et al., 2002*). Chromosome-dependent enrichment of a combination of TFBSs in centromeric and pericentromeric regions is thought to be involved in supporting cell type-dependent centromere clustering. Through Hi-C data analysis for inter-chromosomal interactions in human interphase nuclei, we identified interactions of TFBS islands of different chromosomes (*Wada, Wada & Ikemura, 2020*). Considering the findings associated with the human genome, the large-scale structures discovered in insect genomes may also be involved in the formation of nuclear three-dimensional structures. To clarify these issues, detailed analyses of Hi-C data of insects should be performed.

### Biological significance of genome signature of insects

The biological significance of genome signature and the evolutionary processes that have led to their establishment are of particular interest. Oligonucleotide composition is
clearly related to various biological functions. Regarding dinucleotides, CG is particularly important as the target for C methylation in diverse lineages. In the case of *B. putata* and *A. agestis*, CG composition tends to increase near the chromosome ends (Fig. 7 and Fig. S13) and this feature appears to be similar to that of frogs and bats (*Katsura et al., 2021*; *Iwasaki et al., 2022*). For frogs and bats, there is a distinct increase in the CG/GC ratio and the odds ratio of CG near the chromosome ends, indicating a specific CG increase rather than a G + C% increase. For *B. putata* and *A. agestis*, GC also similarly increase (Fig. S13), reflecting a simple G + C% increase. For vertebrates, a pronounced CG deficiency, which is thought to be related to C methylation, has been reported (*Bogdanovic & Veenstra, 2009*), whereas in insects, the CG deficiency is not as pronounced as in vertebrates. This should be the reason for the differences in CG distribution observed in insects and vertebrates. Although CG-deficiency levels in insects are clearly lower than in vertebrates, *Bewick et al. (2017)* analyzed CG-deficiency levels and CG-methylation levels in a wide phylogenetic range of insects and discovered clear differences between species, indicating relationships with methyltransferases possessed by each species. As the CG methylation is important for epigenomic regulation, the analytical methods presented in Fig. 3 should be useful for better understanding epigenomic regulation. It should also be noted that the large-scale study of oligonucleotide usage in different species using AI, which is capable of analyzing even incompletely sequenced genomes, is a promising area of research.

In the present study, insect species-specific oligonucleotide compositions (genome signatures) were analyzed using unsupervised AI. Our analysis revealed SZ sequences at the 1-Mb and 100-kb levels with oligonucleotide compositions that differ from the genome signatures of each species. SZ sequences identified in the human genome are considered to be functionally important due to the apparent enrichment of TFBS. SZ sequences found in a grasshopper genome also show similar trends to the SZ sequences found in the human genome, suggesting that similar mechanisms for interphase nuclear organization exist in the insect genome.

## CONCLUSIONS

The genome signature is of particular interest in terms of the mechanisms and biological significance that have caused the species-specific differences in oligonucleotide composition, and is considered a powerful search needle to explore the various roles of genome sequences other than protein coding. By analyzing the di-, tri- and pentanucleotide compositions in 100 kb and 1 Mb sequences, BLSOM has clustered most of the sequences by species and thus shown that the unique genome signature extends over almost the entire region of each insect genome. In addition, BLSOM has revealed diagnostic oligonucleotides responsible for this clustering (self-organization). Since BLSOM is an unsupervised clustering method, the clustering of sequences was performed without any information about the species, and therefore, not only the interspecies separation, but also the intraspecies separation can be achieved. A clear example of the intraspecies separation is the Mb-level (GA + TC)-rich regions found at and near the ends of *S. americana* chromosomes. Based on the profound knowledge of the human genome, as

well as the cytogenetic information of the grasshopper chromosomes (*Souza & Melo, 2007*), we predicted that the focused Mb-level regions correspond to centromeric and pericentromeric regions, which are enriched in various TFBS core motifs as previously found for the human genome. This alignment-free method is suitable for comparative genomics between phylogenetically distant species. The number of species whose genomes have only been sequenced and little other experimental research has been conducted is rapidly increasing, and the present alignment-free method can provide a wide range of information for these poorly characterized genomes by using the profound knowledge accumulated even in the distantly related species.

## ACKNOWLEDGEMENTS

We gratefully acknowledge the valuable comments of Dr. Takashi Abe in Niigata University.

### Funding
This work was supported by the Collaborative Research Grant of Nagahama Institute of Bio-Science and Technology. The funders had no role in study design, data collection and analysis, decision to publish, or preparation of the manuscript.

### Grant Disclosures
The following grant information was disclosed by the authors:
Collaborative Research Grant of Nagahama Institute of Bio-Science and Technology.

### Competing Interests
The authors declare there are no competing interests.

### Author Contributions
- Yui Sawada performed the experiments, analyzed the data, prepared figures and/or tables, authored or reviewed drafts of the article, and approved the final draft.
- Ryuhei Minei conceived and designed the experiments, authored or reviewed drafts of the article, and approved the final draft.
- Hiromasa Tabata conceived and designed the experiments, authored or reviewed drafts of the article, and approved the final draft.
- Toshimichi Ikemura performed the experiments, analyzed the data, prepared figures and/or tables, authored or reviewed drafts of the article, and approved the final draft.
- Kennosuke Wada analyzed the data, prepared figures and/or tables, and approved the final draft.
- Yoshiko Wada analyzed the data, prepared figures and/or tables, and approved the final draft.
- Hiroshi Nagata conceived and designed the experiments, authored or reviewed drafts of the article, and approved the final draft.

- Yuki Iwasaki conceived and designed the experiments, performed the experiments, analyzed the data, prepared figures and/or tables, authored or reviewed drafts of the article, and approved the final draft.

## Data Availability

The raw data is available in the Supplemental File.

## Supplemental Information

Supplemental information for this article can be found online at http://dx.doi.org/10.7717/peerj.17025#supplemental-information.

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
