# Peer review of "Unsupervised AI reveals insect species-specific genome signatures"

_PeerJ, doi:10.7717/peerj.17025_

## Round 0.1 · original submission · Major Revisions

Your manuscript has been reviewed by two experts in the field (sorry for the delay; it is the winter vacation season). As you can see from their comments below, one of them (Reviewer 1) is relatively positive while the other (Reviewer 2) is relatively negative. Particularly, Reviewer 2 points out that few novel findings are given in terms of the initial questions asked in the Introduction. Please read their comments carefully and address their points accordingly. Looking forward to your revised manuscript.

·

Basic reporting

Very clear and very easy to read. I congratulate the authors on presenting the material in such a succinct and methodical manner. The literature cited is extensive but not overwhelming. This field goes back a long way, so navigating it can be challenging and the authors have done a great job. I was able to download the raw data and view the code on GitHub. This was very easy and clear. The results are interesting and discussed appropriately without inflation of importance.

Experimental design

The experimental design is standard (at least for those of us in this area). Replication would be straightforward (again, at least for those of us who have done this sort of thing before) given the code and the oligonucleotide compositions provided in the supplementary downloads. The use of the U-matrix is very helpful. It is very interesting to see this technique applied to the increasing plethora of half-completed (in some cases) genomes. SOMs on genome signatures may be coming back into a period of renewed interest, since now we have the material on which to use them and in large quantities. The use of SOMs in this way tended to die out in the last decade as deep sequencing promised more direct ways to make sense of genomes, but it is now apparent that such rapid sequencing frequently leaves genomes in states of partial completeness where alignment is difficult and further analysis is challenging. SOMs thus has a chance to return as way into the interpretation of such data and the authors have shown a great example of exactly how it may be applied.

Validity of the findings

Replication, as previously stated, would be straightforward, I believe. I am confident that this paper presents valid findings.

Additional comments

Please check for consistency of italicization, e.g. line 289 Schistocerca should be italicised as a genus name.
Please make sure that all figure legends are full and consistent with each other, e.g. Figure 8 omits to mention that it contains U-matrices (Figure 5 does, for instance).

Reviewer 2 ·

Basic reporting

- With a few exceptions the language is clear.
- The references given are very focussed on papers published by the authors. Slightly more distantly related work such as the classification of genomes and genomic elements using kmer frequencies, but not using self-organising maps, is not cited.
- The manuscript is mostly self contained. It is a bit unfortunate that e.g. Fig. 1 can only be understood when digging out Table S4.
- The paper is well structures. I think that the introduction promises much more than what the authors present in the results, discussion and conclusions.

Experimental design

The authors of the manuscript have analysed 22 insect genomes with various descriptive methods to identify genome specific signatures. They have cut the genomics sequences into segments of length 1Mb and 100kb. For these segments, der kmer frequencies of small kmers in the range 2-5 bp were determined. The kmer frequencies were then clustered with Kohonen self-organizing maps as described in detail in Abe et al. 2003. Furthermore, kmer densities have been analysed and visualised in various ways.

The main finding is that the kmer frequencies differ in different genomes in such a way that these differences can be identified by self organising maps. When reading the manuscript it becomes clear that the authors have used this method for several taxonomic groups, mainly vertebrates. Now they show that self-organizing maps also work for groups of insects.

I like the capabilities of self-organizing maps and the visualisation of the kmer densities along the chromosomes, but altogether I ask myself what a biologist can learn from these analyses. I understand that all sequence segments that are outliers in terms of kmer composition are potentially interesting in particular if this composition is normally quite conserved in the genome. But I cannot see that this has been used in the present manuscript. The manuscript only shows the potential of a method to find outliers in kmer composition, but does not demonstrate that this approach helps us to find a single interesting loci - except the telomeres.

- The degree of originality is limited, since the method has been used for several groups before.
- In my opinion the authors do not show that the results are highly relevant.
- I would assume that the analyses have been carried out in a rigorous way.
- Altogether, the methods are described well.

Validity of the findings

1) The authors apply a method they have already applied on multiple taxonomic groups. It is the first time this method is applied to insects as far as I can see. Nevertheless the novelty is limited.

2) All data has been provided as far as I can see.

3) Requirement: The conclusions should be appropriately stated, should be connected to the original question investigated, and should be limited to those supported by the results. In particular, claims of a causative relationship should be supported by a well-controlled experimental intervention. Correlation is not causation.

In short:
I do not see that this is given. The main finding is that the authors have found a trend similar to other speculative papers/findings. No question mentioned in the introduction has been answered. It remains unclear, why I should use this method.

Long answer:
In the introduction the authors talk about the advantages of alignment free methods, but the real advantages of the methods used do not become clear to me. The speculative nature becomes obvious in the conclusion: "SZ sequences identified within the human genome are thought to be functionally important due to the apparent enrichment of TFBSs. A similar trend was observed for SZ sequences found in insect genomes, suggesting the presence of comparable mechanisms for interphase nuclear organization of each insect genome. A promising avenue of research should be the large-scale analysis of oligonucleotide usage across species, using AI to analyze large datasets."
I do not see why the authors say that they see a trend and what I could learn from the suggested large scale analysis with these methods? There are few reasons why I should cite this paper since the findings are basically that genomes differ in the kmer composition, which was already known. I think it would be necessary to show that and how truly interesting loci can be found with this method. So far the results is purely descriptive.

After reading the introduction I was looking forward to see examples how the method reveals interesting biological features. After reading the conclusions I was a bit disappointed.

I am sorry that I cannot find more positive words for this manuscript.

Additional comments

Further problems I see:
1) The authors talk about oligonucleotides and oligonucleotide composition. I find this highly confusing, since to my knowledge the only accepted usage for the word oligonucleotides is to refer to real DNA molecules, often synthesised artificially. This is what wikipedia says: "Oligonucleotides are short DNA or RNA molecules, oligomers, that have a wide range of applications in genetic testing, research, and forensics." I see that the authors have used this term in their papers before, a fact that does not fully convince me. If I read "oligonucleotide composition" I think of medical applications of short sequences, but not of kmer frequencies in a genomic sequence. Googling the term confirms my feeling that this is not the generally accepted usage. In bioinformatics the short substrings the authors are talking about are consistently referred to as kmers and I would strongly suggest to do the same in this paper.

2) I do not understand the following sentences in "Results", section starting at lines 174: "The number of grid points (nodes) was set as such that an average of 10 sequences was attributed to each node." Later I read: "In the figure, nodes containing sequences of a single species are shown in species-specific colors, and those containing sequences of multiple species are displayed in black." Now in the Figure most nodes have a colour so the number of species is one. How can it be that the overall average is 10 if only a very small number of nodes are black. The rather few black nodes would need to have a really high number of species, which is not expected.

3) The authors write: "For S. americana, while the entire box for the G+C% and its median were less than 50%, numerous outliers were located above 50%." My first question was, "outlier from what." Going back I see that this sentence refers to regions in the genome that have a CG content higher than 50%. I think a more direct language would be helpful.

4) If one wants to understand Fig. 1, it is necessary to dig out table S4 to find the taxa associated with the colours. Well, I see that the space on figures is valuable and the main message of the paper can only be brought to the reader with figures.

---

## Round 0.2 · accepted · Accept

Your revised manuscript was reviewed by one of the two original reviewers, who now recommends its acceptance. Unfortunately, the other reviewer, who was rather negative about the original manuscript, did not accept my invitation to re-review. Because of the reviewer's opinion on the novelty of scientific papers, inferred from his/her comments, I am doubtful if he/she is satisfied with this revision. However, I myself confirmed that the authors have made reasonable efforts to address his/her criticisms and have rewritten the manuscript largely. Therefore, I now recommend its acceptance to the section editor. Congratulations!

·

Basic reporting

All issues resolved.

Experimental design

All issues resolved.

Validity of the findings

All issues resolved.

Additional comments

This is now ready to publish.